# Blue Photons from Broad-Spectrum LEDs Control Growth, Morphology, and Coloration of Indoor Hydroponic Red-Leaf Lettuce

**DOI:** 10.3390/plants12051127

**Published:** 2023-03-02

**Authors:** Qingwu Meng, Erik S. Runkle

**Affiliations:** 1Department of Plant and Soil Sciences, University of Delaware, 531 South College Avenue, Newark, DE 19716, USA; 2Department of Horticulture, Michigan State University, 1066 Bogue Street, East Lansing, MI 48824, USA

**Keywords:** indoor vertical farming, green light, red light, sole-source lighting, white light

## Abstract

For indoor crop production, blue + red light-emitting diodes (LEDs) have high photosynthetic efficacy but create pink or purple hues unsuitable for workers to inspect crops. Adding green light to blue + red light forms a broad spectrum (white light), which is created by: phosphor-converted blue LEDs that cast photons with longer wavelengths, or a combination of blue, green, and red LEDs. A broad spectrum typically has a lower energy efficiency than dichromatic blue + red light but increases color rendering and creates a visually pleasing work environment. Lettuce growth depends on the interactions of blue and green light, but it is not clear how phosphor-converted broad spectra, with or without supplemental blue and red light, influence crop growth and quality. We grew red-leaf lettuce ‘Rouxai’ in an indoor deep-flow hydroponic system at 22 °C air temperature and ambient CO_2_. Upon germination, plants received six LED treatments delivering different blue fractions (from 7% to 35%) but the same total photon flux density (400 to 799 nm) of 180 μmol·m^−2^·s^−1^ under a 20 h photoperiod. The six LED treatments were: (1) warm white (WW_180_); (2) mint white (MW_180_); (3) MW_100_ + blue_10_ + red_70_; (4) blue_20_ + green_60_ + red_100_; (5) MW_100_ + blue_50_ + red_30_; and (6) blue_60_ + green_60_ + red_60_. Subscripts denote photon flux densities in μmol·m^−2^·s^−1^. Treatments 3 and 4 had similar blue, green, and red photon flux densities, as did treatments 5 and 6. At the harvest of mature plants, lettuce biomass, morphology, and color were similar under WW_180_ and MW_180_, which had different green and red fractions but similar blue fractions. As the blue fraction in broad spectra increased, shoot fresh mass, shoot dry mass, leaf number, leaf size, and plant diameter generally decreased and red leaf coloration intensified. Compared to blue + green + red LEDs, white LEDs supplemented with blue + red LEDs had similar effects on lettuce when they delivered similar blue, green, and red photon flux densities. We conclude that the blue photon flux density in broad spectra predominantly controls lettuce biomass, morphology, and coloration.

## 1. Introduction

Photons with wavelengths between 400 and 750 nm are essential to plant growth and development by driving photosynthetic activity, regulating morphological adaption, and modulating secondary metabolism. This wavelength range is typically divided into four 100 nm wavebands: blue (400 to 499 nm), green (500 to 599 nm), red (600 to 699 nm), and far red (700 to 750 nm). Photosynthesis is wavelength-dependent as these photons carry energy that differentially excites two photosystems in plant cell membranes in light-dependent reactions of photosynthesis [1,2]. Although the traditionally defined photosynthetically active radiation (400 to 700 nm) only includes blue, green, and red light, recent research has shown the equal efficacy of supplemental far-red light in whole-plant photosynthesis [2,3,4]. Moreover, light mediates plant morphological responses by controlling photoreceptors. Blue light is absorbed by photoreceptors including cryptochrome 1, cryptochrome 2, and phototropins [5], whereas red and far-red light are primarily absorbed by members in the phytochrome family [6]. Both cryptochromes and phytochromes can absorb green light, albeit poorly [7]. Increasing the fraction of blue photons relative to green or red photons generally decreases indoor-grown lettuce extension growth and biomass accumulation but increases pigmentation, whereas increasing the fraction of far-red photons generally does the opposite by eliciting the shade avoidance response [8,9,10].

In the last decade, lighting for controlled-environment agriculture has evolved rapidly thanks to the improved energy efficiency, spectral tuning, and design of light-emitting diode (LED) fixtures. These improvements increased the commercial adoption of LED fixtures, especially in indoor vertical farms. Growers can choose from a wide range of LED fixtures with different specifications. The selection depends, in part, on the fixture cost, photosynthetic photon efficacy (the photosynthetic photon output per unit energy in μmol∙J^−1^), photon spectrum, form factor, and light responses of crops [11]. Although blue + red LEDs are prevalent in horticultural applications primarily because of their high photon and photosynthetic efficacy [11,12], there are merits to broad-spectrum LEDs. The addition of green light to blue + red light creates white light to the human eye but decreases the photosynthetic photon efficacy [13]. Unlike blue + red LEDs, which create pink or purple hues, broad-spectrum LEDs can reveal the true colors of plants, which facilitates workers’ inspection of crop growth, nutrient conditions, insects, and diseases in a visually pleasing environment. High color fidelity under broad-spectrum LED lighting can be especially desirable in indoor vertical farms.

Although combining monochromatic blue, green, and red LEDs can create white light, commercial white LEDs are typically blue LEDs with a phosphor coating that converts most of the blue photons to photons at longer wavelengths (i.e., green, red, and far red), thereby emitting a broad range of biologically active radiation [13]. Depending on the material and amount of phosphor coatings, white LEDs create different hues, as indicated by the correlated color temperature (CCT). With a low percentage of blue light (e.g., 7%), warm-white LEDs have a low CCT of 2500 to 3500 K, whereas neutral-white (3500 to 4500 K), cool-white (4500 to 5500 K), and daylight (5500 to 7500 K) LEDs have higher percentages of blue light (e.g., 20% and 30%, respectively) and higher photon efficacies [11,14]. These white LEDs also differ in how they reveal the true colors of plants, as indicated by the color rendering index (CRI, negative to 100) or the TM-30 fidelity index (*R*_f_, 0 to 100), with higher indices indicating higher color fidelity. Other specialized white LEDs also exist, such as mint-white (also known as equalized-white) LEDs. Developed by OSRAM Opto Semiconductors, these are blue LEDs with an efficient green phosphor coating that can be combined with red LEDs to create high-CRI white light [15].

Horticultural lighting companies often supplement white LEDs with monochromatic blue and/or red LEDs to create distinct broad spectra. First, this allows for spectrum customization to elicit specific plant responses (e.g., high yield, compact growth, and increased production of secondary metabolites) [16]. Second, including more efficient blue (peak wavelength ≈ 450 nm) and/or red (peak wavelength ≈ 660 nm) LEDs increases fixture photosynthetic photon efficacy while still providing acceptable color rendering at a low cost [15]. From an energy efficiency standpoint, warm-white (CCT = 2700 K) and cool-white (CCT = 6500 K) LEDs have lower photon efficacies (2.6 and 2.9 μmol·J^−1^, respectively) than blue, red, and far-red LEDs (3.5, 4.5, and 4.7 μmol·J^−1^, respectively) [11]. However, the overall performance of LED fixtures and light use efficiency of plants depend on both photosynthetic photon efficacy and crop responses under different photon spectra. While it is straightforward to compare cost and efficacy of lighting fixtures, there is little information about how indoor leafy greens grow under phosphor-converted broad spectra, with and without supplemental blue (peak wavelength ≈ 450 nm) + red (peak wavelength ≈ 660 nm) LEDs, relative to monochromatic blue + green (peak wavelength ≈ 525 nm) + red LEDs. Therefore, the objective of this study was to characterize the growth responses of hydroponic red-leaf lettuce (*Lactuca sativa*) ‘Rouxai’ under sole-source LED lighting with various broad spectra, sometimes with matching waveband-integrated photon flux densities. We hypothesized the broad spectra with higher ratios of blue to green or red light would lead to lower lettuce biomass, decreased extension growth, and greater red leaf coloration.

## 2. Results

### 2.1. Biomass and Morphology

Lettuce shoot fresh mass was greatest under WW_180_, MW_180_, and B_20_G_60_R_100_, and lowest under MW_100_B_50_R_30_ and B_60_G_60_R_60_ (Figure 1). Shoot dry mass showed a similar response as shoot fresh mass. Plants grew similarly under WW_180_ and MW_180_ with 78–98% and 39–62% higher shoot fresh and dry mass, respectively, than those that grew similarly under MW_100_B_50_R_30_ and B_60_G_60_R_60_. Compared to WW_180_, shoot fresh and dry mass was 14% lower under MW_100_B_10_R_70_, but similar under B_20_G_60_R_100_. Plants developed approximately three fewer leaves when grown under MW_100_B_50_R_30_ and B_60_G_60_R_60_ than the other treatments. Plant biomass corresponded to leaf and plant size; the treatment trends were similar for shoot dry mass and plant diameter. Likewise, the leaves were longest and widest for the plants grown under WW_180_ and MW_180_ and shortest and narrowest for the plants grown under MW_100_B_50_R_30_ and B_60_G_60_R_60_. Leaf length and width were 17–19% and 25–35% greater, respectively, under WW_180_ and MW_180_ than MW_100_B_50_R_30_ and B_60_G_60_R_60_.

### 2.2. Coloration

Plants grown under WW_180_ and MW_180_ were visually greener with little red coloration, whereas those grown under the other treatments had distinct foliage redness (Figure 2). As shown by *L** (lower is darker; higher is brighter), leaf brightness was higher under WW_180_ and MW_180_ than B_20_G_60_R_100_, MW_100_B_50_R_30_, and B_60_G_60_R_60_. As shown by *a** (lower is greener; higher is redder), leaves were greenest under WW_180_ and MW_180_ and reddest under MW_100_B_50_R_30_, and B_60_G_60_R_60_. As shown by *b** (lower is bluer; higher is yellower), leaves were yellower under WW_180_ and MW_180_ than the other treatments. The relative chlorophyll index (SPAD value) was 12–16% lower for plants grown under MW_180_ than the other treatments.

### 2.3. Phenotypes Influenced by Blue Light

The six treatments in this study provided a wide range of blue photon flux densities, from 12 to 62 μmol∙m^−2^∙s^−1^. Several major lettuce phenotypic parameters showed dose-dependent relationships to the blue photon flux density (Figure 3). Increasing the blue photon flux density linearly decreased the shoot fresh and dry mass, leaf length and width, and plant diameter while increasing the leaf redness (as shown by *a**).

## 3. Discussion

In this study, indoor hydroponic red-leaf lettuce ‘Rouxai’ grew similarly under warm-white and mint-white LEDs, which provided similar blue fractions of 7% and 9%, respectively, but different green, red, and far-red fractions (Table 1). Low blue light was a signal to increase extension growth of lettuce ‘Rouxai’, irrespective of green and red light, which led to greater shoot growth [8]. Warm-white LEDs emitted more far-red photons (10%) than mint-white LEDs (3%), leading to red-to-far-red ratios of 5.3 and 8.1, respectively, and internal phytochrome photoequilibria of 0.661 and 0.722, respectively (Table 1). Despite these differences in spectra, warm-white and mint-white LEDs had comparable effects on lettuce growth (biomass), morphology (leaf size), and coloration. Lettuce shoot biomass was greatest under these two white LED types at least partly due to greater leaf expansion and canopy size, which increased light interception for photosynthesis. However, lettuce appeared mostly green under the two white spectra and lacked red coloration, indicating low anthocyanin accumulation. Because high-energy blue photons induce rapid anthocyanin accumulation and enhance plant resilience in crops including red-leaf lettuce [17,18,19], the low blue fractions in the two white spectra could explain the lack of red leaf coloration. In this study, lettuce grown under broad spectra with higher blue photon flux densities had redder leaves. Similar to our results, partially substituting equalized-white light with blue light increased red leaf coloration and anthocyanin concentration of red-leaf lettuce ‘Red Butter’ [20].

The effects of 44% substitution of mint-white light with blue + red light on lettuce growth depended on the blue-to-red ratio. The lower substitute blue-to-red ratio of 1:7 changed the blue, green, red, and far-red photon flux densities by 2.6, −49.4, 47.8, and −2.3 μmol∙m^−2^∙s^−1^ (17%, −46%, 94%, and −37%), respectively (Table 1). At a blue photon flux density of 16–18 μmol∙m^−2^∙s^−1^, this substitution of green photons with red photons decreased lettuce shoot fresh mass by 12% (but not dry mass), decreased leaf length by 8%, intensified red leaf coloration, and increased chlorophyll concentration. In a separate study with a similar blue photon flux density, substituting 60 μmol∙m^−2^∙s^−1^ of green photons with red photons also decreased lettuce shoot fresh mass by 15% and leaf number but did not influence other parameters [8]. On the other hand, partial substitution of mint-white light with the higher blue-to-red ratio (5:3) changed the blue, green, red, and far-red photon flux densities by 43.1, −48.9, 7.9, and −2.6 μmol∙m^−2^∙s^−1^ (275%, −45%, 16%, and −42%), respectively (Table 1). This increased the blue photon flux density, which decreased lettuce shoot fresh and dry mass by 44% and 28%, respectively; decreased leaf number, leaf size, and canopy size; and increased red leaf coloration and chlorophyll concentration. Similarly, substituting green photons with blue photons decreased shoot mass and leaf expansion while increasing coloration and chlorophyll concentration of lettuce ‘Rex’ and ‘Rouxai’ and kale (*Brassica oleracea* var. *sabellica*) ‘Siberian’ [9].

We combined monochromatic blue + green + red LEDs to deliver similar 100 nm waveband flux densities of blue, green, red, and far-red light to the two mint-white + blue + red treatments. The former created three distinct peaks of narrowband radiation, whereas the latter created somewhat broader, more continuous photon distributions from the phosphor conversion of blue LEDs. Despite these spectral distribution differences, all measured lettuce phenotypes were similar when the integrated blue, green, red, and far-red photon wavebands were similar. However, narrowband green LEDs have a low efficacy (0.54 μmol∙J^−1^ at full power), whereas mint-white LEDs emit green photons at a much higher efficacy (1.52 μmol∙J^−1^ at full power) [15]. In addition, the color fidelity of plants was superior under mint-white + blue + red LEDs (CRI = 77–82) than their blue + green + red counterparts (CRI = 58–61) or mint-white LEDs alone (CRI = 63). The higher visual quality under mint-white + blue + red LEDs creates better working conditions and facilitates plant inspection for nutrient and physiological disorders and integrated pest management. Similarly, mint-white + red LEDs (3:1 or 9:11) delivered a higher CRI of 72–77 than blue + green + red LEDs (CRI = 56) or mint-white LEDs alone (CRI = 63) [15]. Lastly, warm-white LEDs are more ubiquitous and have a nearly perfect CRI of 97 but have a slightly lower efficacy of (1.28 μmol∙J^−1^ at full power) than mint-white LEDs, although both are considered white light.

We identified a dose-dependent inverse relationship between the blue photon flux density in a broad spectrum and the measured lettuce phenotypes in this study (Figure 3). The blue photon flux density appears to accurately predict these lettuce phenotypes, although these models could be further strengthened in future research by including additional broadband spectra with intermediate blue photon flux densities, especially between 25 and 60 μmol∙m^−2^∙s^−1^. In general, a broad spectrum with a higher blue photon flux density (in the range of 12–62 μmol∙m^−2^∙s^−1^) progressively decreased lettuce shoot biomass, leaf size, and canopy size but increased red leaf coloration. In a previous study, increasing the blue photon flux density in blue + green + red LEDs from 0 to 100 μmol∙m^−2^∙s^−1^ at a fixed photosynthetic photon flux density (PPFD) of 180 μmol∙m^−2^∙s^−1^ produced similar results [8]. In contrast, increasing the blue photon flux density from 22 to 55 μmol∙m^−2^∙s^−1^ in a broad spectrum (from blue + green + red or warm-, neutral-, or cool-white LEDs) at a PPFD of 200 μmol∙m^−2^∙s^−1^ decreased dry mass and leaf area index of tomato (*Solanum lycopersicum*), but not other crops including green-leaf lettuce ‘Waldmann’s Green’ [21]. The four broad-spectrum treatments in that study had similarly high green fractions (41–48% of the PPFD), whereas the green fractions in this study and [8] were 32–33% of the PPFD in all treatments except for the mint-white LED treatment, in which 62% of the PPFD was of green light. The discrepancy in blue light responses could be at least partly attributed to potential blue and green light interactions. The inconsistent responses among studies could also be caused by different plant densities and maturities at harvest.

Although the effects of blue light on plant growth and morphology vary by species and spectral conditions such as the PPFD [22], increasing blue light in combined blue + red LEDs generally restricts lettuce extension growth, light capture, and biomass accumulation [23,24,25]. Cryptochromes 1 and 2 are photoreceptors that absorb blue light and regulate extension growth [26]. In arabidopsis (*Arabidopsis thaliana*), PHYTOCHROME-INTERACTING FACTOR 4 (PIF4) and PIF5 are transcription factors that promote extension growth in the shade avoidance response [27]. Under low blue light, cryptochromes 1 and 2 interact with PIF4 and PIF5 to promote extension growth [28]. However, under sufficiently high blue light, cryptochromes 1 and 2 suppress the function of PIF4 while cryptochrome 2 and PIF5 are targeted for degradation [28]. A similar mechanism in lettuce could explain the reduced extension growth under broad spectra containing high blue light.

## 4. Materials and Methods

### 4.1. Plant Material and Propagation

We performed this experiment twice over time as a randomized complete block design in the Controlled-Environment Lighting Laboratory at Michigan State University (East Lansing, MI, USA). We rinsed and soaked rockwool cubes (AO 25/40, 25 × 25 × 40 mm; Grodan, Milton, ON, Canada) with deionized water, drained excess water from plastic trays, sowed one seed of red-leaf lettuce ‘Rouxai’ (Johnny’s Selected Seeds, Winslow, ME, USA) per cube, and covered them with transparent humidity domes. Seeds germinated under warm-white LED fixtures (CCT = 2700 K, PHYTOFY RL; OSRAM, Beverley, MA, USA) at a total photon flux density (400–799 nm) of 50 μmol∙m^−2^∙s^−1^ during the first 24 h at an air temperature setpoint of 20 °C. Based on visual assessment, we achieved a uniform seed germination rate of >95%. We then grew seedlings under 180 μmol∙m^−2^∙s^−1^ of warm-white light under a 20 h photoperiod at an air temperature setpoint of 22 °C until transplant on day 13. A total photon flux density of 180 μmol∙m^−2^∙s^−1^ was within the typical range delivered in commercial indoor vertical farms, while a 20 h photoperiod led to a daily light integral of 13 mol∙m^−2^∙d^−1^, which sufficed lettuce seedling growth [29]. The seedling trays sat on top of foam boards floating in tubs filled with water, in which the seedlings were subsequently transplanted.

After removing the humidity domes on day 4, we began lighting treatments and sub-irrigated the rockwool cubes as needed daily to submerge one fourth of the cube height in a nutrient solution with a pH of 5.7–5.9 and electrical conductivity of 1.2–1.4 dS∙m^−1^. We prepared it in 18.9-L buckets by dissolving a 12.0N–1.7P–13.3K base fertilizer (12–4–16 RO Hydro FeED; JR Peters, Inc., Allentown, PA, USA) and magnesium sulfate (Epsom salt; Pennington Seed, Inc., Madison, GA, USA) sequentially in deionized water. The nutrient solution provided seedlings with the following nutrients (in mg∙L^−1^): 125 N, 18 P, 139 K, 73 Ca, 49 Mg, 39 S, 1.7 Fe, 0.52 Mn, 0.56 Zn, 0.13 B, 0.47 Cu, and 0.13 Mo. We adjusted the nutrient solution pH with dry potassium bicarbonate and diluted (1:31) 95% to 98% sulfuric acid (J.Y. Baker, Inc., Phillipsburg, NJ, USA).

### 4.2. Lighting Treatments

On day 4, we transferred uniform lettuce seedlings to six broad-spectrum treatments. These treatments delivered the same total photon flux density of 180 μmol∙m^−2^∙s^−1^ under a 20 h photoperiod (0200–2200 h), resulting in a daily light integral of 13 mol∙m^−2^∙d^−1^, which was sufficient to produce lettuce in the exponential growth phase [29,30]. Tunable multi-colored LED fixtures (PHYTOFY RL; OSRAM) delivered warm-white (WW_180_) light, mint-white (MW_180_) light, mint-white partially substituted with two ratios of blue + red light (MW_100_B_10_R_70_ and MW_100_B_50_R_30_), and combined monochromatic blue + green + red light with two ratios of blue + red light (B_20_G_60_R_100_ and B_60_G_60_R_60_). The number after each LED type indicates its photon flux density in μmol∙m^−2^∙s^−1^. During the setup of each lighting treatment, we took spectral scans at nine representative locations at plant canopy (46 cm below the LED fixtures) with a spectroradiometer (PS200; Apogee Instruments, Inc., Logan, UT, USA). We used the average photon flux density to adjust the fixture output in lighting control software (Spartan Control Software 2018 version 1; OSRAM) until the average was ±3 μmol∙m^−2^∙s^−1^ for each LED type. The spectral characteristics and distributions of the six lighting treatments are shown in Table 1 and Figure 4, respectively. The peak wavelengths of warm-white, mint-white, blue, green, and red LEDs were 639, 559, 449, 526, and 664 nm, respectively.

### 4.3. Hydroponic System and Environment

On day 13, we transplanted lettuce seedlings into the 36-cell foam rafts (60.9 × 121.9 × 2.5 cm; Beaver Plastics, Ltd.; Acheson, AB, Canada) floating on top of nutrient solutions in flood tables (1.22 × 0.61 × 0.18 m; Active Aqua AAHR24W; Hydrofarm, Petaluma, CA, USA) of a deep flow technique hydroponic system. At a planting density of 48.4 plants∙m^−2^, plant centers were 20.3 cm apart horizontally and 14.6 cm apart diagonally. The nutrient solutions were constantly recirculated with water pumps and aerated with air stone discs (20.3 × 2.5 cm; Active Aqua AS8RD; Hydrofarm), which were submerged in three reservoirs and connected to external air pumps (Active Aqua AAPA70L; Hydrofarm). The nutrient solutions contained the same two-part fertilizers in deionized water as for seedlings and provided the following initial nutrients at transplant (in mg∙L^−1^): 150 N, 22 P, 166 K, 88 Ca, 58 Mg, 47 S, 2.1 Fe, 0.63 Mn, 0.68 Zn, 0.15 B, 0.56 Cu, and 0.15 Mo.

From transplant to harvest, we measured the nutrient solution pH, electrical conductivity, and temperature daily with a pH and electrical conductivity meter (HI9814; Hanna Instruments, Woonsocket, RI, USA). Each of these parameters was similar among the three reservoirs, which were used for the six lighting treatments, throughout the two replications (Figure 5). Whenever the pH was <5.1, we used potassium bicarbonate to raise the nutrient solution pH to 5.6–5.9 until day 28, after which we did not adjust it to observe the effects of plant nutrient uptake. Because 98% of the total nitrogen in the base fertilizer was in the nitrate form, maturing lettuce increased the nutrient solution pH. We replenished three reservoirs with deionized water periodically to ensure the water pumps were fully submerged in the reservoirs but did not provide additional fertilizers. Consequently, the nutrient solution electrical conductivity decreased gradually from 1.8 to 1.9 dS∙m^−1^ at transplant to 1.5 to 1.7 dS∙m^−1^ at harvest.

A ventilation and air-conditioning unit (HBH030A3C20CRS; Heat Controller, LLC., Jackson, MI, USA), which was connected to a wireless thermostat controller (Honeywell International, Inc., Morris Plains, NJ, USA), maintained a constant air temperature setpoint of 22 °C. Plants were grown at the ambient CO_2_ concentration. We monitored the environment with a temperature and relative humidity sensor (HMP110; Vaisala, Inc., Louisville, CO, USA) and a CO_2_ sensor (GMD20; Vaisala, Inc.). Sensors were wired to a datalogger (CR1000; Campbell Scientific, Inc., Logan, UT, USA) that recorded hourly means of 10 s intervals (Figure 6). The mean air temperature, CO_2_ concentration, and relative humidity (mean ± sd) were 22.4 ± 0.6 °C, 410 ± 50 μmol∙mol^−1^, and 34% ± 10%, respectively, in replication 1 and 22.5 ± 0.6 °C, 398 ± 35 μmol∙mol^−1^, and 35% ± 7%, respectively, in replication 2.

### 4.4. Data Collection and Analysis

On day 30 and 33 in the two consecutive replications, we harvested and conducted destructive measurements on eight randomly selected plants from each lighting treatment. For each plant, we measured shoot fresh mass with a top-loading balance (GX-1000; A&D Store, Inc., Wood Dale, IL, USA), the number of leaves longer than 3 cm, plant diameter (the longest horizontal distance between plant edges), and the length and width of the sixth most mature true leaf. At three random locations on recently matured leaves of each plant, we also measured the International Commission on Illumination *L***a***b** color space values with a color reader (Chroma Meter CR-400; Konica Minolta Sensing, Inc., Chiyoda, Tokyo, Japan) and the relative chlorophyll index (SPAD value) with a chlorophyll meter (SPAD-502; Konica Minolta Sensing, Inc.). These pigmentation measurements targeted unshaded interveinal leaf tissue. Subsequently, we placed each plant in a paper bag, dried it for ≥5 d at 60 °C in a forced air drying oven (Blue M, Blue Island, IL, USA), and measured shoot dry mass. In addition, we photographed a representative plant from each lighting treatment from overhead to document crop appearance (Figure 1 and Figure 2). We analyzed plant data with SAS (version 9.4; SAS Institute, Inc., Cary, NC, USA) using the PROC MEANS, PROC MIXED, and PROC GLIMMIX procedures and Tukey’s honestly significant difference test (*α* = 0.05). We also performed regression analysis between the blue photon flux density and plant parameters with Microsoft Excel (Microsoft, Redmond, WA, USA). All data were pooled for analysis because of the non-significance of the treatment × replication interaction term (*p* > 0.05) and/or similar treatment trends between replications.

## 5. Conclusions

Warm-white and mint-white LEDs with distinctly different broad spectra had similar effects on indoor hydroponic lettuce biomass, morphological traits, and pigmentation. When a phosphor-converted broad spectrum (mint white) was partially substituted with blue + red light, the change in the blue photon flux density primarily determined the plant phenotypic responses. Lettuce grew similarly under mint-white LEDs partially substituted with blue + red LEDs and blue + green + red LEDs that delivered similar photon flux densities of 100 nm blue, green, and red wavebands. Lastly, the blue photon flux density in the six broad spectra tested was an accurate predictor of lettuce growth and coloration.

## Figures and Tables

**Figure 1 plants-12-01127-f001:**
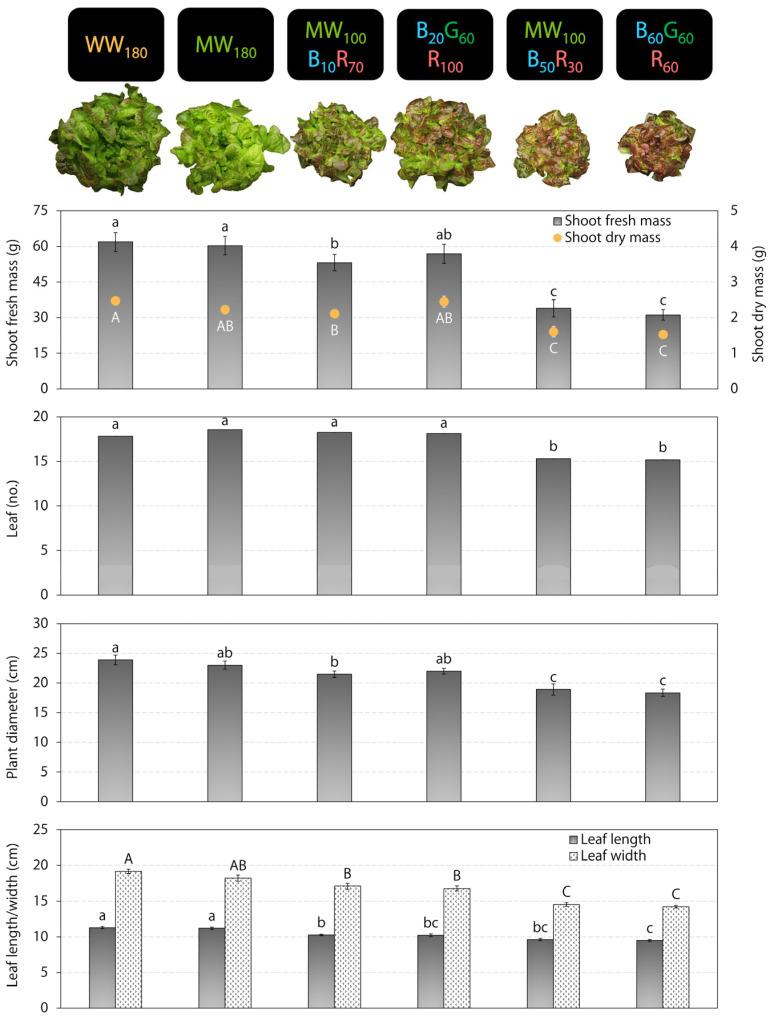
Shoot fresh and dry mass, leaf number, plant diameter, and length and width of the sixth most mature true leaf of red-leaf lettuce ‘Rouxai’ grown under six indoor lighting treatments delivered by warm-white (WW), mint-white (MW), blue (B), green (G), and/or red (R) light-emitting diodes (LEDs) (*n* = 20). The number after each LED type is its photon flux density in μmol∙m^−2^∙s^−1^. For each parameter, values followed by different letters are statistically different (*α* = 0.05).

**Figure 2 plants-12-01127-f002:**
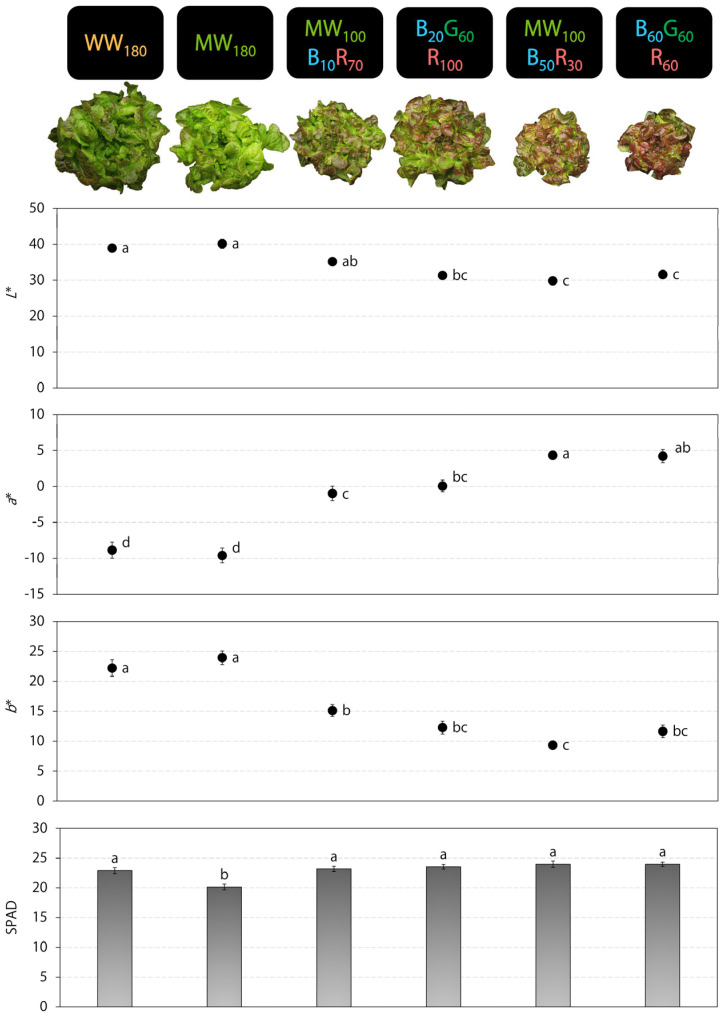
Leaf color indices (*L***a***b**) and relative chlorophyll index (SPAD value) of red-leaf lettuce ‘Rouxai’ grown under six indoor lighting treatments delivered by warm-white (WW), mint-white (MW), blue (B), green (G), and/or red (R) light-emitting diodes (LEDs) (*n* = 20). The number after each LED type is its photon flux density in μmol∙m^−2^∙s^−1^. For each parameter, values followed by different letters are statistically different (*α* = 0.05).

**Figure 3 plants-12-01127-f003:**
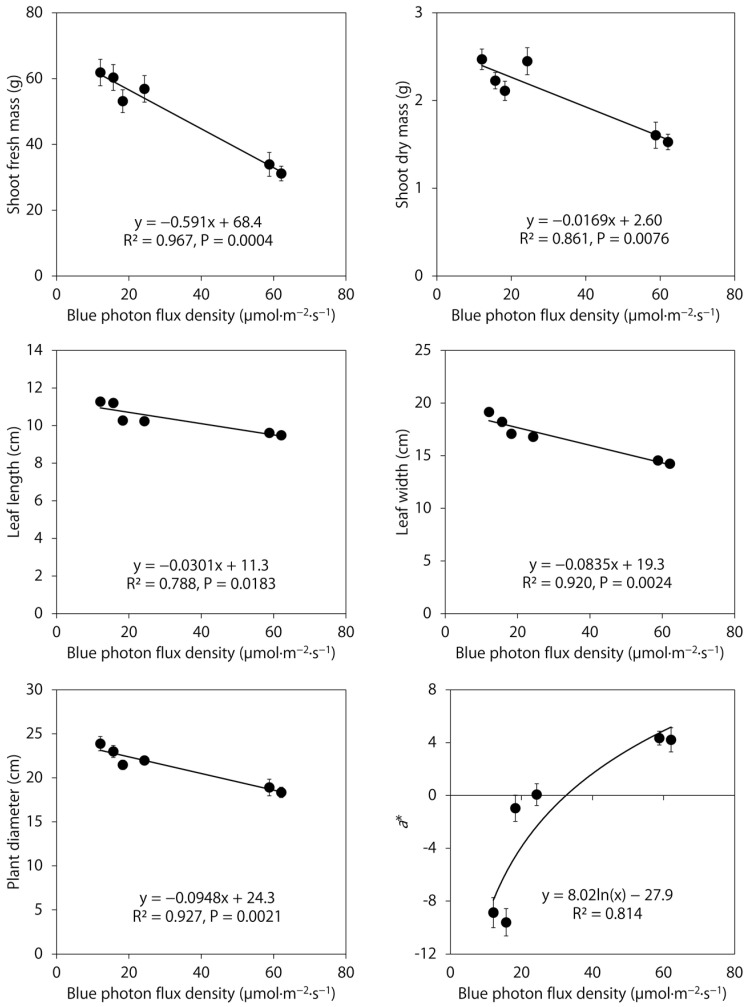
Relationships between the blue photon flux density and shoot biomass, morphology, and red leaf coloration parameters of red-leaf lettuce ‘Rouxai’ grown under six indoor lighting treatments delivered by warm-white, mint-white, blue, green, and/or red light-emitting diodes, which provided a range of blue photon flux densities (*n* = 20).

**Figure 4 plants-12-01127-f004:**
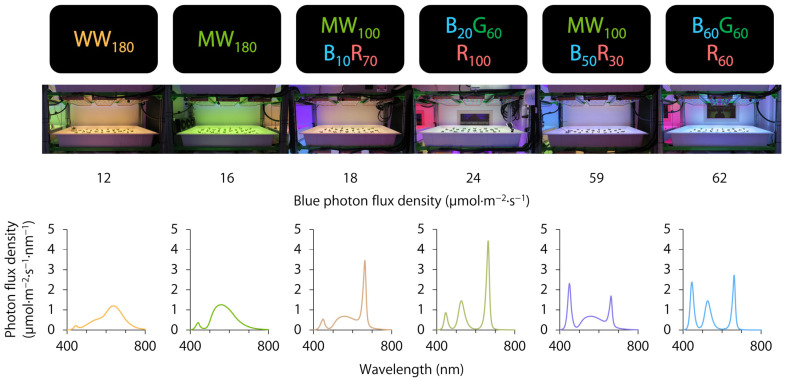
Photos and spectral distributions of six indoor lighting treatments delivered by warm-white (WW), mint-white (MW), blue (B), green (G), and/or red (R) light-emitting diodes (LEDs). The number after each LED type is its photon flux density in μmol∙m^−2^∙s^−1^.

**Figure 5 plants-12-01127-f005:**
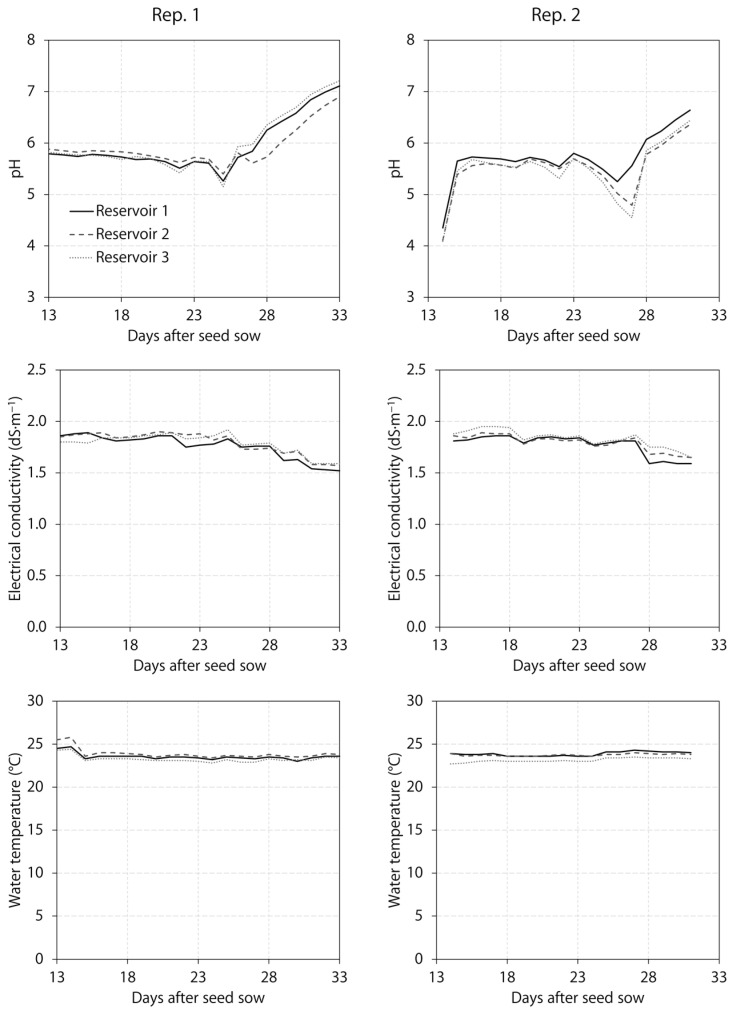
Nutrient solution pH, electrical conductivity, and temperature of three reservoirs used throughout two experimental replications (Rep.) for six indoor lighting treatments delivered by warm-white, mint-white, blue, green, and/or red light-emitting diodes (LEDs). Reservoir 1 was for the two blue + green + red LED treatments. Reservoir 2 was for the mint-white LED treatment. Reservoir 3 was for the warm-white LED treatment and the two mint-white + blue + red LED treatments.

**Figure 6 plants-12-01127-f006:**
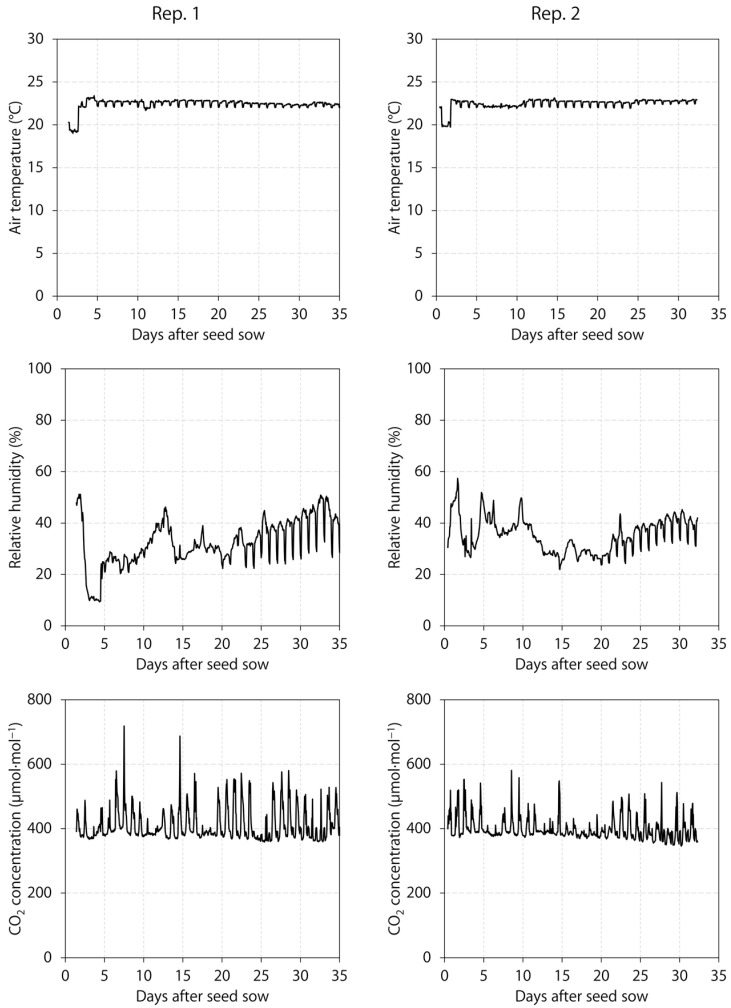
Air temperature, relative humidity, and CO_2_ concentration throughout two experimental replications (Rep.) of the growth room housing six indoor lighting treatments delivered by warm-white, mint-white, blue, green, and/or red light-emitting diodes. The CO_2_ peaks were caused by people inside the growth room.

**Table 1 plants-12-01127-t001:** Spectral characteristics of six broad-spectrum lighting treatments delivered by warm-white (WW), mint-white (MW), blue (B), green (G), and/or red (R) light-emitting diodes (LEDs).

Lighting Treatment	WW_180_ ^1^	MW_180_	MW_100_B_10_R_70_	B_20_G_60_R_100_	MW_100_B_50_R_30_	B_60_G_60_R_60_
Photon flux density and percentage of each waveband (μmol∙m^−2^∙s^−1^)						
B ^2^	12.1 (7%)	15.7 (9%)	18.3 (10%)	24.3 (13%)	58.8 (33%)	62.1(35%)
G	51.9 (29%)	107.7 (60%)	58.3 (33%)	58.9 (32%)	58.8 (33%)	58.8 (33%)
R	98.4 (54%)	50.6 (28%)	98.4 (55%)	99.1 (54%)	58.5 (32%)	57.5 (32%)
FR	18.6 (10%)	6.2 (3%)	3.9 (2%)	1.2 (1%)	3.6 (2%)	0.7 (0%)
Integrated photon flux density (μmol∙m^−2^∙s^−1^)						
PPFD ^3^	162.4	174.0	175.0	182.3	176.1	178.4
TPFD	181.0	180.2	179.0	183.5	179.7	179.1
YPFD ^4^	148.9	154.1	156.8	156.5	150.1	146.1
Light ratios and metrics						
B:R	0.12	0.31	0.19	0.25	1.01	1.08
R:FR	5.3	8.1	25.0	85.5	16.3	84.6
Ippe ^4^	0.661	0.722	0.814	0.843	0.771	0.813
CRI ^5^	97	63	82	58	77	61

^1^ The number after each LED type is its photon flux density in μmol∙m^−2^∙s^−1^. ^2^ Each 100 nm waveband is defined as follows: B, 400–499 nm; G, 500–599 nm; R, 600–699 nm; and far red (FR), 700–799 nm. ^3^ The photosynthetic photon flux density (PPFD, 400–699 nm) and total photon flux density (TPFD, 400–799 nm) differed in the inclusion of far-red (FR) light. ^4^ The yield photon flux density (YPFD, 300–799 nm) and internal phytochrome photoequilibrium (iPPE) were calculated following [31,32], respectively. ^5^ The color rendering index (CRI) was calculated with the OSRAM Sylvania LED ColorCalculator.

## Data Availability

The datasets used and/or analyzed in this manuscript are available from the first corresponding author on reasonable request by e-mail.

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
