# Peer review of "Blue Photons from Broad-Spectrum LEDs Control Growth, Morphology, and Coloration of Indoor Hydroponic Red-Leaf Lettuce"

_plants, 2023, doi:10.3390/plants12051127_

Round 1
Reviewer 1 Report
The authors studied the effect of six broad-spectrum light treatments on growth, developmental, and colorimetric characteristics of red lettuce. Specifically, the manuscript mainly focuses on the portion of blue light, while the discussion also revolves around far-red, red, and green spectral zones. In the introduction, the authors thoroughly explained why they selected these particular light fixtures. The objectives are clear. However, since some readers may not be familiar with the subject, I suggest including a paragraph about the effects of the most prominent spectra (i.e. far-red, red, blue, possibly green), especially on lettuce for which literature is vast.
The Materials and methods clearly describe the experimental procedure. The data/results in Figures and Tables are clear. I enjoyed the short but on-point description of the results.
Overall, the manuscript provides valuable information about the effects of light spectra (especially blue light) for red lettuce production.
Specific comments are following.
· L96, L111. From your experience, is 20-h photoperiod optimal for lettuce growth? How did you select this photoperiod, instead of 18-h for example?
· Figure 1 could be enlarged to fit the page width, in order to be easily readable.
Author Response
Response to Reviewer 1 Comments
The authors studied the effect of six broad-spectrum light treatments on growth, developmental, and colorimetric characteristics of red lettuce. Specifically, the manuscript mainly focuses on the portion of blue light, while the discussion also revolves around far-red, red, and green spectral zones. In the introduction, the authors thoroughly explained why they selected these particular light fixtures. The objectives are clear. However, since some readers may not be familiar with the subject, I suggest including a paragraph about the effects of the most prominent spectra (i.e. far-red, red, blue, possibly green), especially on lettuce for which literature is vast.
Response: We appreciate the feedback and have added a paragraph at the beginning of the Introduction section to provide an overview of spectral effects on plants, especially lettuce.
The Materials and methods clearly describe the experimental procedure. The data/results in Figures and Tables are clear. I enjoyed the short but on-point description of the results.
Overall, the manuscript provides valuable information about the effects of light spectra (especially blue light) for red lettuce production.
Response: We appreciate the kind feedback.
Specific comments are following.
- L96, L111. From your experience, is 20-h photoperiod optimal for lettuce growth? How did you select this photoperiod, instead of 18-h for example?
Response: Based on previous studies, at a total photon flux density of 180 µmol∙m–2∙s–1, a moderately long photoperiod of 20 h resulted in a daily light integral of 13 mol∙m–2∙d–1, which was sufficient for lettuce growth in both the seedling and exponential growth phases. We have provided this justification with citations in both L96 and L111.
- Figure 1 could be enlarged to fit the page width, in order to be easily readable.
Response: We have enlarged Figure 1 to fit the page width.
Reviewer 2 Report
69-70, 79 etc. needed precise information about wavelength in supplementation color of light e.g. green, blue, red
92 can be introduce method of evaluation progress in germination (e.g.by phptpgrametric measurement during different stages of germination
263, 346 if possible useful more details e.g. the plant biomass, surface etc.
Your very promissing research-developing study could be in the futiure supplemented by SEM and X-ray analysis of essential trace elements accumulation in different groups of plants in relation to algorithms of fotostimulation by LEDs if possible related also to comparative study using e.g. delay luminescence versus growth rate, etc.
Author Response
Response to Reviewer 2 Comments
69-70, 79 etc. needed precise information about wavelength in supplementation color of light e.g. green, blue, red
Response: We have added peak wavelength information of blue, green, and red LEDs in L69-70 and L79.
92 can be introduce method of evaluation progress in germination (e.g.by phptpgrametric measurement during different stages of germination
Response: We have added the following sentence in L95 to clarify the method of germination evaluation: “Based on visual assessment, we achieved a uniform seed germination rate of >95%.” Because seed germination occurred in a uniform manner, it satisfied the experimental requirement of uniform seedlings. Thus, we did not need to quantify germination progress at different stages.
263, 346 if possible useful more details e.g. the plant biomass, surface etc.
Response: In L263, we have specified “growth” as biomass and morphology as “leaf size”. Because we have elaborated details of plant biomass and leaf size in the Results section, we would prefer focusing the Discussion and Conclusions on overall key findings.
Your very promissing research-developing study could be in the futiure supplemented by SEM and X-ray analysis of essential trace elements accumulation in different groups of plants in relation to algorithms of fotostimulation by LEDs if possible related also to comparative study using e.g. delay luminescence versus growth rate, etc.
Response: We appreciate the kind feedback. We will keep this suggestion in mind for future research.